# Impact of COVID-19 on Intracranial Meningioma Resection: Results from California State Inpatient Database

**DOI:** 10.3390/cancers14194785

**Published:** 2022-09-30

**Authors:** Muni Rubens, Anshul Saxena, Venkataraghavan Ramamoorthy, Md Ashfaq Ahmed, Zhenwei Zhang, Peter McGranaghan, Emir Veledar, Michael McDermott

**Affiliations:** 1Miami Cancer Institute, Baptist Health South Florida, Miami, FL 33176, USA; 2Center for Advanced Analytics, Baptist Health South Florida, Miami, FL 33143, USA; 3Herbert Wertheim College of Medicine, Florida International University, 11200 SW 8th St., Miami, FL 33199, USA; 4Department of Internal Medicine and Cardiology, Charité-Universitätsmedizin Berlin, Corporate Member of Freie Universität Berlin and Humboldt Universität zu Berlin, Augustenburger Platz 1, 10117 Berlin, Germany; 5Miami Neuroscience Institute, Baptist Health South Florida, 8950 N Kendall Dr. Suite 407W, Miami, FL 33176, USA

**Keywords:** coronavirus, mortality, hospitalization, intracranial meningioma resection, morbidity

## Abstract

**Simple Summary:**

All fields of healthcare were adversely affected by the COVID-19 pandemic. In this study, we sought to understand the effects of COVID-19 on hospitalizations for intracranial meningioma resection using a large database. We compared hospitalization rates as well as hospital outcomes such as Clavien–Dindo grade IV complications, in-hospital mortality, and prolonged length of stay for intracranial meningioma resection during 2019 and 2020. Our findings showed that though hospitalization rates decreased slightly during the COVID-19 pandemic, hospital outcomes were not adversely affected. The findings of our study show that with adequate planning and preparations, better hospital outcomes could be sustained even during healthcare emergencies such as COVID-19 pandemic. Our findings assure that neurosurgery practice in the US ensured the best quality of care to their patients even during COVID-19 pandemic.

**Abstract:**

Purpose: To assess the effects of COVID-19 on hospitalizations for intracranial meningioma resection using a large database. Methods: We conducted a retrospective analysis of the California State Inpatient Database (SID) 2019 and 2020. All adult (18 years or older) hospitalizations were included for the analysis. The primary outcomes were trends in hospitalization for intracranial meningioma resection between 2019 and 2020. Secondary outcomes were Clavien–Dindo grade IV complications, in-hospital mortality, and prolonged length of stay, which was defined as length of stay ≥75 percentile. Results: There were 3,173,333 and 2,866,161 hospitalizations in 2019 and 2020, respectively (relative decrease, 9.7%), of which 921 and 788 underwent intracranial meningioma resection (relative decrease, 14.4%). In 2020, there were 94,114 admissions for COVID-19 treatment. Logistic regression analysis showed that year in which intracranial meningioma resection was performed did not show significant association with Clavien–Dindo grade IV complications and in-hospital mortality (OR, 1.23, 95% CI: 0.78–1.94) and prolonged length of stay (OR, 1.05, 95% CI: 0.84–1.32). Conclusion: Our findings show that neurosurgery practice in the US successfully adapted to the unforeseen challenges posed by COVD-19 and ensured the best quality of care to the patients.

## 1. Introduction

All fields of healthcare have been severely affected by the COVID-19 pandemic. Neurosurgery is no exception, and elective surgeries have been extensively affected resulting in significantly lower admission rates, and postponements or even cancellations of surgical procedures [1,2]. This was due to the diversion of both healthcare resources and personnel for COVID-19 management, so that both staff and equipment could be directed towards COVID-19 related critical care needs [3,4]. The implementation of control measures such as isolation and quarantine could have additionally decreased the admission rates of neurosurgical cases [5,6]. In addition, many patients opted to willfully delay treatment for the fear of COVID-19 contraction during hospital encounters [7]. Many healthcare providers have also experienced unforeseen challenges such as working in fields beyond their expertise, managing cases with scarce resources, overcoming ethical issues, and risking their own lives to COVID-19 [8,9,10,11].

Intracranial meningiomas are the most common type of benign primary intracranial tumors [12]. They represent almost 37% of all primary central nervous system tumors and 50% of all benign brain tumors [12]. The majority of them are asymptomatic and are often detected as incidental findings on magnetic resonance imaging [13]. During the period 2004 to 2017, the incidence of meningioma showed increasing trends. However, after this period, the incidence plateaued and decreased [14]. Interventions are required when these tumors grow in size within a short period of time and become symptomatic [15]. Management of meningioma is primarily surgical resection and aims towards maximal or complete removal of tumor and its dural tail [16,17]. These procedures are usually performed electively and were affected by the COVID-19 pandemic.

In this study, we intended to understand the effects of COVID-19 on hospitalizations for intracranial meningioma resection using a large database. Specifically, we compared trends in monthly hospitalization rates for intracranial meningioma resection between 2019 and 2020. In addition, we also compared trends monthly hospitalization rates for intracranial meningioma resection and COVID-19 hospitalizations in 2020. Furthermore, we explored whether there were differences in adverse in-hospital outcomes after intracranial meningioma resection between the two years. The results of our study could provide a detailed understanding of the effects of COVID-19 on intracranial meningioma resection in California and could give population-level estimates that could be useful to the healthcare system.

## 2. Materials and Methods

### 2.1. Study Design and Data Source

The current study was a retrospective analysis of data retrieved from the California State Inpatient Database (SID) gathered during 2019 and 2020. The SID was developed by the Agency for Healthcare Research and Quality (AHRQ) for collecting statewide inpatient clinical data [18]. SID has data from patients admitted to the participating hospitals within the state [18]. Every year, discharge data from >90% of patients admitted to the participating hospitals are collected and stored by the SID. We used the Strengthening the Reporting of Observational Studies in Epidemiology (STROBE) guideline for assuring the quality of the study [19].

### 2.2. Study Population

All adult (18 years or older) hospitalizations for intracranial meningioma resection that occurred during 2019 and 2020 were included for the analysis. To identify these hospitalizations, we used the International Classification of Diseases, Tenth Revision, Clinical Modification (ICD-10-CM) diagnosis code D32.0 for benign neoplasms of the cerebral meninges, and procedural codes 00510ZZ, 00B10ZZ, 00C10ZZ, and 00D10ZZ for intracranial meningioma resection.

### 2.3. Study Variables and Outcomes

The primary outcomes of the study were trends in hospitalization for intracranial meningioma resection between 2019 and 2020. Secondary outcomes were Clavien–Dindo grade IV complications, in-hospital mortality, and prolonged length of stay. Prolonged length of stay was defined as length of stay ≥75 percentile for the entire population. Clavien–Dindo grade IV complications include life-threatening complications due to dysfunction one or more organ systems and requiring management in intensive care units [20]. The Clavien–Dindo grade IV complications include components such as severe sepsis or septic shock, acute renal failure requiring hemodialysis, pulmonary embolism, acute myocardial infarction, cardiac arrest requiring cardiopulmonary resuscitation, prolonged mechanical ventilation, and unplanned intubation or re-intubation for managing postoperative respiratory failure. We identified these components using ICD-10 codes (Appendix A). The Clavien–Dindo grade IV complications have been previously used in studies using administrative data [21,22,23]. Other variables included age, sex, race, insurance status, clinical risk profile, and Elixhauser comorbidity index.

### 2.4. Statistical Analysis

Descriptive statistics were used to understand the differences in demographic and clinical risk profile between hospitalization for intracranial meningioma resection that occurred during 2019 and 2020. Categorical variables were described as frequencies and percentages and compared using chi-square test. Monthly trends in hospitalization for intracranial meningioma resection between 2019 and 2020 and COVID-19 hospitalizations during 2020 were calculated and graphically plotted against one another. Logistic regressions were used to find the differences in adverse clinical outcomes such as Clavien–Dindo grade IV complications and in-hospital mortality, and prolonged length of stay during 2019 and 2020. In our models, we adjusted for covariates such as age, sex, race, insurance status, and clinical risk profiles such as hypertension, diabetes mellitus, obesity, coagulation disorder, peripheral vascular disease, liver disease, chronic renal failure, alcohol abuse, and drug abuse. Sensitivity analysis was done among hospitalization of older adults aged 65 years and above. In this analysis we used logistic regressions to find the differences in Clavien–Dindo grade IV complications and in-hospital mortality during 2019 and 2020. Statistical significance was set at *p* < 0.05 and all tests were two-sided. All statistical analyses were conducted using SAS, version 9.4 (SAS Inc., Cary, NC, USA).

## 3. Results

There were 3,173,333 and 2,866,161 hospitalizations in 2019 and 2020, respectively (relative decrease, 9.7%). Among these hospitalizations, 921 and 788 underwent intracranial meningioma resection in 2019 and 2020, respectively (relative decrease, 14.4%). Among all hospitalizations in 2020, there were 94,114 admissions for COVID-19 treatment. Comparison of demographic and clinical characteristics of hospitalizations for intracranial meningioma resection that occurred in 2019 and 2020 showed that for both of the years the majority of the patients were in the age group 45–64 years (45.2% versus 43.9%) and majority were females (70.2% versus 69.5%). During both years, the majority of the hospitalizations occurred among Whites (51.0% versus 47.6%), followed by Hispanics (22.6% versus 23.9%), Asian or Pacific Islander and Native American (14.1% versus 15.3%), and Blacks (6.7% versus 5.4%). Insurance status across both the years showed that majority had private insurance coverages (42.1% versus 41.6%), followed by Medicare (38.4% versus 38.2%), and Medicaid (15.3% versus 17.3%). The most common comorbidity was hypertension (51.2% versus 51.4%), followed by obesity (19.7% versus 19.4%), diabetes mellitus (10.0% versus 10.0%). Majority had Elixhauser comorbidity index values of 1 or 2 during both years (46.3% versus 43.7%). None of the demographic and clinical characteristics differed between 2019 and 2020. Table 1 shows the demographic and clinical characteristics of hospitalizations for intracranial meningioma resection during 2019 and 2020.

A comparison of clinical outcomes of hospitalizations for intracranial meningioma resection between 2019 and 2020 showed that Clavien–Dindo grade IV complications such as severe sepsis or septic shock (1.4% versus 1.6%, *p* = 0.688), prolonged requirement of mechanical ventilation (3.3% versus 4.3%, *p* = 0.251), and grade IV complication (4.5% versus 5.3%, *p* = 0.399) did not differ significantly across the two years. Likewise, prolonged length of stay (27.0% versus 27.9%, *p* = 0.683) did not differ significantly between 2019 and 2020. Table 2 shows the clinical outcomes of hospitalizations for intracranial meningioma resection during 2019 and 2020.

Trends of hospitalizations for intracranial meningioma resection during 2019 and 2020 showed that the rates of these hospitalizations were generally lower in 2020, except for February, May, and November. Trends of hospitalizations for intracranial meningioma resection during 2020 showed an unusual dip during April, which also corresponded with a spike in COVID-19 hospitalizations. Similarly, the decrease observed during July 2020 corresponds with the spike in COVID-19 hospitalizations. The rates of hospitalizations started to increase from August to October, corresponding with decreases in COVID-19 hospitalizations. Eventually, there was a steep decline in hospitalizations from November to December, corresponding with an exponential increase in COVID-19 hospitalizations. Figure 1 shows the trends in hospitalizations for intracranial meningioma resection during 2019 and 2020 and COVID-19 hospitalizations during 2020.

Logistic regression analysis showed that year in which intracranial meningioma resection was performed (OR, 1.23, 95% CI: 0.78–1.94) did not show significant association with Clavien–Dindo grade IV complications and in-hospital mortality. The odds of Clavien–Dindo grade IV complications and in-hospital mortality were significantly higher among Blacks (OR, 3.56, 95% CI: 1.65–7.65) and those with chronic renal failure (OR, 5.52, 95% CI: 2.84–10.76). Table 3 shows the odds ratios for factors associated with Clavien–Dindo grade IV complications and in-hospital mortality.

Similarly, regression analysis showed that the year in which intracranial meningioma resection was performed (OR, 1.05, 95% CI: 0.84–1.32) did not show significant association with prolonged length of stay. The odds of prolonged length of stay were significantly higher among those with hypertension (OR, 1.59, 95% CI: 1.23–2.06), coagulation disorder (OR, 4.24, 95% CI: 2.67–6.73), and chronic renal failure (OR, 1.85, 95% CI: 1.19–2.88). Table 4 shows the odds ratios for factors associated with prolonged length of stay.

Sensitivity analysis done among hospitalizations ≥65 years showed that year was not significantly associated with Clavien–Dindo grade IV complications and in-hospital mortality (OR, 1.27, 95% CI: 0.64–2.50). Appendix A shows the odds ratios for factors associated with Clavien–Dindo grade IV complications and in-hospital mortality among hospitalizations ≥65 years.

## 4. Discussion

To the best of our knowledge, this is the first study that explored the impact of COVID-19 hospitalizations for intracranial meningioma resection. Using California SID, we found that hospitalizations for intracranial meningioma resection were in general lower during 2020, compared to 2019. However, there were no significant differences in adverse clinical outcomes across the years. We found that variations in hospitalizations for intracranial meningioma resection during 2020 corresponded with the fluctuations in rates of COVID-19 hospitalizations.

The decrease in hospitalizations for resection of intracranial meningiomas in our study is consistent with observations in other studies which show that a number of COVID-19 related factors could be responsible for these results [24,25,26]. The majority of these admissions could have been delayed because people were scared of contracting COVID-19 and were willing to postpone them [7,27,28]. COVID-19 measures such as shelter-in-place and substantial prioritization of healthcare resources for managing COVID-19 could have further decreased healthcare delivery for conditions other than COVID-19 [3,29]. The Centers for Medicare & Medicaid Services and US Surgeon General recommended urgent ban and postponing of all elective surgeries in the US during March 2020. Studies have also shown that the rates of neurosurgeries were substantially affected by COVID-19 pandemic. For example, in a study among two academic neurosurgery centers in New Orleans, in 2020, there was a 34% decrease in monthly neurosurgical volume [30]. Similarly, a study done at an academic neurosurgery department reported that admission rates for neurosurgeries decreased by 69% during COVID-19 pandemic [31]. In addition, the COVID-19 pandemic has caused considerable disruption to the production and supply chains as well as trading and transportation routes [32,33]. This could have precipitated shortages of required logistics for delivering these procedures and deliberate postponements.

In our study, the rates of hospitalizations for resection of intracranial meningiomas substantially decreased during April 2020. Similar decreases were also reported in other studies during the initial spike of the pandemic. For example, in a study done among 8 neurosurgical training programs, the mean number of cases decreased by 49% during April 2020 [34]. Similarly, a study performed in a neurosurgery department at the Johns Hopkins University School of Medicine reported 45% decline in inpatient census during the same period [31]. Elective operations were compulsorily prohibited in some institutions during March and April when the pandemic was at its peak [30,31]. Reallocation of operating rooms for accommodating COVID-19 assigned care areas could have precipitated these findings [30]. In addition, factors such as cancellation of events and stay-at-home mandate issued by the government of California in March 2020 could have caused the substantial decline observed in our study [25]. Subsequently, the reopening plans proposed by the governor of California was followed by a gradual increase in the number of these hospitalizations until November 2020, when it exceeded the pre-COVID-19 levels [35]. Reintroduction of COVID-19 prevention measures during November and December, when the states experienced a raging wave of the pandemic was followed by severe plummeting of hospitalizations for resection of intracranial meningiomas observed during the end of 2020.

We found that in-hospital clinical outcomes following intracranial meningioma resection such as Clavien–Dindo grade IV complications and in-hospital mortality and prolonged length of stay were not different between 2019 and 2020. Clavien–Dindo grade IV complications include single or multiple organ system failures requiring intensive care unit admissions and interventions [20]. Similar findings were also observed in previous studies where COVID-19 did not have significant impact on adverse outcomes such as mortality and prolonged length of stay as well as perioperative complications [30,31,36]. These findings show that although the volume of hospitalizations for intracranial meningioma resection have decreased due to diversion of resources for managing the pandemic, healthcare providers have strived to provide the highest quality of care.

We also found that Clavien–Dindo grade IV complications and in-hospital mortality were significantly higher among Blacks, indicating racial disparities in in-hospital outcomes following intracranial meningioma resection. Previous studies have shown that Black patients had significantly worse outcomes following management for intracranial meningioma. For example, a study done by Anzalone et al. using surveillance, epidemiology, and end results (SEER) database showed that Black patients had the worst disease specific and overall survival rates at 5 years, compared to other races [37]. A systematic review that included 55 studies showed that in general Black patients had higher rates of complications and in-hospital mortality following neurosurgical treatments for intracranial tumors [38]. Understanding the reasons for these disparities and trying to overcome them through effective interventions could significantly help in alleviating them and achieving equity in healthcare. In addition, we found that comorbidities such as chronic renal failure was associated with Clavien–Dindo grade IV complications and in-hospital mortality, and hypertension, coagulation disorder, and chronic renal failure with prolonged length of stay following hospitalization for intracranial meningioma resection. These comorbidities should be adequately scrutinized and controlled before planning for surgical treatment for this condition. Such measures could significantly improve hospital outcomes and quality of life among meningioma patients.

### Strengths and Limitations

We used California SID for our analysis. California had the largest number of COVID-19 cases within the US. Therefore, we could accurately estimate the impact of COVID-19 on relatively rare hospitalizations such as those for intracranial meningioma resection. In addition, due to the larger sample size, our results have greater external validity.

One of the main limitations of this study was that we used an administrative database. All variables in this study were retrieved from billing codes assigned to the conditions and procedures following patients’ discharge. There could be some inaccuracies during coding, leading to misclassification bias. These inconsistencies could have affected our findings. The results are limited to the duration of hospitalization because SID does not have information prior to or after hospitalization. Therefore, the complications that developed after discharge could not be ascertained. SID does not have information on grading and staging of meningioma, or other specifics such as tumor size, location, tumor extension, extent of extraction, surgical methods, and other non-surgical managements such as radiotherapy and embolization. Some meningioma pathologies require urgent hospitalization. However, we could not ascertain these pathologies because SID is an administrative database. In addition, provider details such as experience and expertise of the neurosurgeon, and surgical techniques used at different hospitals were not available. The availability of these important cofounders could have substantially improved our estimates.

## 5. Conclusions

Our study showed that hospitalizations for intracranial meningioma resection were generally lower during 2020, compared to pre-COVID-19 levels. There were fluctuations in hospitalization trends which corresponded with changes in COVID-19 hospitalizations. Despite additional burden on healthcare systems due to COVID-19 pandemic, there were no significant differences in adverse clinical outcomes. These findings indicate that healthcare providers strived to deliver optimal levels of care in spite of the challenges posed by the pandemic. These findings also assure that neurosurgery practice in the US successfully adapted to these unforeseen challenges and ensured the best quality of care to their patients.

## Figures and Tables

**Figure 1 cancers-14-04785-f001:**
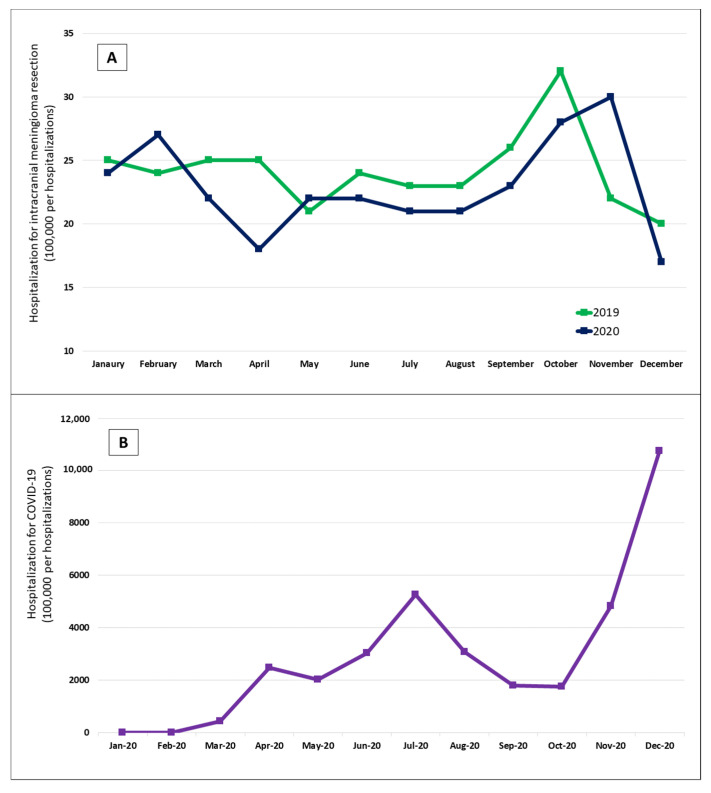
Trends in hospitalizations for intracranial meningioma resection during 2019 and 2020 (**A**) and COVID-19 hospitalizations during 2020 (**B**).

**Table 1 cancers-14-04785-t001:** Demographic and clinical characteristics of hospitalizations for intracranial meningioma resection during 2019 and 2020.

Characteristic	2019 n = 921 (53.9%)	2020 n = 788 (46.1%)	*p* Value
Age, n (%)			0.858
18–44 years	139 (15.1%)	124 (15.7%)	
45–64 years	416 (45.2%)	346 (43.9%)	
≥65 years	366 (39.7%)	318 (40.4%)	
Sex, n (%)			0.750
Male	274 (29.8%)	240 (30.5%)	
Female	647 (70.2%)	548 (69.5%)	
Race/ethnicity, n (%)			0.216
White	459 (51.0%)	372 (47.6%)	
Black	60 (6.7%)	42 (5.4%)	
Hispanic	203 (22.6%)	187 (23.9%)	
Asian or Pacific Islander and Native American	127 (14.1%)	120 (15.3%)	
Other	51 (5.7%)	61 (7.8%)	
Insurance status, n (%)			0.427
Medicare	354 (38.4%)	301 (38.2%)	
Medicaid	141 (15.3%)	136 (17.3%)	
Private insurance	388 (42.1%)	327 (41.6%)	
Other	38 (4.1%)	23 (2.9%)	
Clinical risk profile, n (%)			
Hypertension	472 (51.2%)	405 (51.4%)	0.951
Diabetes mellitus	92 (10.0%)	79 (10.0%)	0.980
Obesity	181 (19.7%)	153 (19.4%)	0.902
Coagulation disorder	50 (5.4%)	41 (5.2%)	0.835
Peripheral vascular disease	47 (5.1%)	43 (5.5%)	0.744
Liver disease	17 (1.8%)	26 (3.3%)	0.055
Chronic renal failure	53 (5.8%)	51 (6.5%)	0.536
Alcohol abuse	13 (1.4%)	11 (1.3%)	0.798
Drug abuse	27 (2.9%)	16 (2.0%)	0.235
Elixhauser comorbidity index, n (%)			0.482
0	197 (21.4%)	169 (21.4%)	
1 or 2	426 (46.3%)	344 (43.7%)	
≥3	298 (32.4%)	275 (34.9%)	

**Table 2 cancers-14-04785-t002:** Clinical outcomes of hospitalizations for intracranial meningioma resection during 2019 and 2020.

Characteristic	2019 n = 1165 (66.8%)	2020 n = 575 (33.2%)	*p* Value
Clavien–Dindo grade IV complications			
Severe sepsis or septic shock	13 (1.4%)	13 (1.6%)	0.688
Acute renal failure requiring dialysis	NR	NR	---
Pulmonary embolism	NR	NR	---
Acute myocardial infarction or cardiac arrest requiring cardiopulmonary resuscitation	NR	NR	---
Prolonged requirement of mechanical ventilation	30 (3.3%)	34 (4.3%)	0.251
Unplanned intubation/reintubation	NR	NR	---
Any grade IV complication	41 (4.5%)	42 (5.3%)	0.399
In-hospital mortality	NR	12 (1.5%)	---
Prolonged length of stay, n (%)	249 (27.0%)	220 (27.9%)	0.683

Note: NR is not reported to comply with Healthcare Cost and Utilization Project data use agreement guideline of not to report tabulated data in a cell size ≤ 10 to protect individual identification.

**Table 3 cancers-14-04785-t003:** Factors associated with Clavien–Dindo grade IV complications and in-hospital mortality.

Characteristic	Odds Ratio (95% CI)
Year

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

**Table 4 cancers-14-04785-t004:** Factors associated with prolonged length of stay.

Characteristic	Odds Ratio (95% CI)
Year

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

## Data Availability

Data are available publicly for purchase at: https://www.hcup-us.ahrq.gov/sidoverview.jsp, accessed on 12 February 2022.

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
