# Peer review of "Impact of COVID-19 on Intracranial Meningioma Resection: Results from California State Inpatient Database"

_cancers, 2022, doi:10.3390/cancers14194785_

Round 1
Reviewer 1 Report
This is a study evaluating the impact of the COIVD pandemic on meningioma surgery in California. The authors used an administrative database to evaluate whether there were differences regarding patient outcome after meningioma surgery. The authors did no detect differences in major complications or lenghth of hospital stay while noting that case numbers declined during 2020.
COVID cases started to rise beginning at the end of February/ start of March 2020. What was the rationale of using 01-12/2020 and would it not be more useful to compare March 2020-March 2021 with the year before that?
The authors state that reasons for decrease in caseload could be fear of contracting COVID in the hospital. Is there data whether meningioma patients did actually contract COVID and how many? Were there severe cases/ mortality related to COVID?
Many countries issued a ban on elective surgery during the pandemic; was there such a ban in place in California? If yes how did this impact the results?
Author Response
Comments: COVID cases started to rise beginning at the end of February/ start of March 2020. What was the rationale of using 01-12/2020 and would it not be more useful to compare March 2020-March 2021 with the year before that?
Reply: Thank you for your comment. We included January to December since first case of COVD-19 was detected in California in January 2020.
https://publichealth.sccgov.org/county-santa-clara-public-health-department-reports-first-case-novel-new-coronavirus
Comments: The authors state that reasons for decrease in caseload could be fear of contracting COVID in the hospital. Is there data whether meningioma patients did actually contract COVID and how many? Were there severe cases/ mortality related to COVID?
Reply: Thank you for your comment. It is a very important question. However, in our sample of meningioma patients none had COVID-19.
Comments: Many countries issued a ban on elective surgery during the pandemic; was there such a ban in place in California? If yes how did this impact the results?
Reply: Thank you for your comment. Centers for Medicare & Medicaid Services and US Surgeon General recommended urgent ban and postponing of all elective surgeries in the US during March 2020. This could have affected meningioma hospitalizations in our sample. We have included that in the Discussion section. All changes are highlighted in red.
Reviewer 2 Report
The manuscript is clear, relevant for the field and presented in a well-structured manner. It is sufficiently scientifically sound.
The introduction has sufficient background and references, the cited references are relevant for the paper, the design of the evaluation is appropriate and adequately described, the results are clearly
presented, and the conclusions are supported by the results.
The figures/tables are appropriates, and properly show the data.
The Authors performed a statistical analysis on the effects of COVID pandemic on intracranial meningiomas surgery, comparing patients that underwent surgical resection in 2019 to patients in 2020, during pandemic.
They analyzed retrospectively the California State Inpatient Database (SID) to evaluate the trends in hospitalizations, compared these trends to the one from COVID pandemic and assessed the effects of pandemic on clinical complications, based on the Clavien-Dindo grade IV, in-hospital mortality,
and length of stay (> 75 percentile).
The hospitalization for intracranial meningioma resection were in general lower during 2020 compared to 2019 (14.4%). Variations in hospitalization corresponded with fluctuation in rates of Covid-19 hospitalization, due to factors attributable on one hand to the hospital organization (as lower admission rate, cancellations of surgery, saturated intensive care unit), and on the other side to the patients (affected by COVID, refusing surgery because scared to get COVID during hospitalization, or that postponed a radiological exam for less accessible hospital, for fear or for restrictive control measures).
No substantial changes, in terms of clinical outcome, mortality and time of hospitalization, were found between patients treated during COVID pandemic in 2020, and patients treated the year before (2019, without COVID pandemic).
- Logistic regression analysis to assess differences in Clavien-Dindo grade IV, in-hospital mortality and prolonged length of staying, was adjusted for some covariates, as age, risk factors, but not for Simpson grade.
The topic of the paper does not concern grading or staging of the tumor or surgical management.
Nevertheless we believe that the analysis should be adjusted also for Simpson grade, which itself can affect the clinical outcome and the in-hospital staying and mortality.
The findings of the study of Rubens and coll. show that, with adequate planning and preparations, better hospital outcomes can be sustained even during healthcare emergencies such as Covid-19 pandemic. The data will be useful to the healthcare system.
Furthermore, these findings reassure that neurosurgery practice in the US successfully adapted to these unforeseen challenges and ensured the best quality of care to their patients.
Question to the authors:
- It would be interesting to know whether the chronic renal failure, as the factor mostly responsible for mortality and worse clinical outcome in intracranial surgery, was likewise observed in patients affected by COVID.
- Furthermore, prolonged mechanical ventilation was a bit more required in 2020 (+1%), although not statistically significant. In this “sub-group”, there was any patient affected by COVID?
- any patient that underwent surgery for intracranial meningioma, resulted affected by COVID during hospitalization?
- The hospital organization system in California, during pandemic, reported any changes, as adopted by Lombardy, the most affected Italian region, in an Hub and Spoke system for Subarachnoid Aneurysmal Hemorrhage, which allowed centralization of neurosurgical emergencies to decrease a
diagnostic delay? [Fiorindi et al. Aneurysmal subarachnoid hemorrhage during the COVID-19 outbreak in a Hub and Spoke system: observational multi center cohort study in Lombardy, Italy. Acta Neurochirurgica (2022) 164. 141-150]
- The authors did not mention a possible strategy to reduce the preoperative waiting list, considering that some pathologies require a faster hospitalizations (as for vascular and malignant brain tumors).
Author Response
We thank the reviewers for their insightful comments and the time taken to review our manuscript. We have carefully considered the reviewers’ comments and responded in a point-by-point fashion, which can be found in the table below. We appreciate the opportunity to utilize the expertise of our peer reviewers and hope this revision will make the article acceptable for publication in Cancers.
Comments: It would be interesting to know whether the chronic renal failure, as the factor mostly responsible for mortality and worse clinical outcome in intracranial surgery, was likewise observed in patients affected by COVID.
Reply: Thank you for your comment. Chronic renal failure was the most significant factor responsible for worse outcomes among hospitalized meningioma patients in our study (Odds ratio, 3.45, 95% CI: 1.57-4.60). However, in our sample of meningioma patients none had COVID-19. Hence, we could not ascertain whether CRF had similar effects on COVID-19 meningioma patients.
Comments: Furthermore, prolonged mechanical ventilation was a bit more required in 2020 (+1%), although not statistically significant. In this “sub-group”, there was any patient affected by COVID?
Reply: Thank you for your comment. In our sample of meningioma patients none had COVID-19.
Comments: Any patient that underwent surgery for intracranial meningioma, resulted affected by COVID during hospitalization?
Reply: Thank you for your comment. In our sample of meningioma patients none had COVID-19.
Comments: The hospital organization system in California, during pandemic, reported any changes, as adopted by Lombardy, the most affected Italian region, in an Hub and Spoke system for Subarachnoid Aneurysmal Hemorrhage, which allowed centralization of neurosurgical emergencies to decrease a diagnostic delay? [Fiorindi et al. Aneurysmal subarachnoid hemorrhage during the COVID-19 outbreak in a Hub and Spoke system: observational multi center cohort study in Lombardy, Italy. Acta Neurochirurgica (2022) 164. 141-150]
Reply: Thank you for your comment. We could not ascertain whether California state had any planned intervention for addressing centralization of neurosurgical emergencies to decrease a diagnostic delay as the dataset we used is administrative.
Comments: The authors did not mention a possible strategy to reduce the preoperative waiting list, considering that some pathologies require a faster hospitalizations (as for vascular and malignant brain tumors).
Reply: Thank you for your comment. We agree that some meningioma pathologies require a faster hospitalization. However, we could not ascertain this because we used administrative data for our study. Since this is an important point, we have included this lack of data in our Limitation section.